# Enhancing Agricultural Image Segmentation with an Agricultural Segment Anything Model Adapter

**DOI:** 10.3390/s23187884

**Published:** 2023-09-14

**Authors:** Yaqin Li, Dandan Wang, Cao Yuan, Hao Li, Jing Hu

**Affiliations:** School of Mathematics and Computer Science, Wuhan Polytechnic University, Wuhan 430024, China; leeyaqin@whpu.edu.cn (Y.L.); wangdandan2299@163.com (D.W.); yc@whpu.edu.cn (C.Y.); lihao@whpu.edu.cn (H.L.)

**Keywords:** image segmentation, adapters, agricultural image segmentation, zero-shot segmentation

## Abstract

The Segment Anything Model (SAM) is a versatile image segmentation model that enables zero-shot segmentation of various objects in any image using prompts, including bounding boxes, points, texts, and more. However, studies have shown that the SAM performs poorly in agricultural tasks like crop disease segmentation and pest segmentation. To address this issue, the agricultural SAM adapter (ASA) is proposed, which incorporates agricultural domain expertise into the segmentation model through a simple but effective adapter technique. By leveraging the distinctive characteristics of agricultural image segmentation and suitable user prompts, the model enables zero-shot segmentation, providing a new approach for zero-sample image segmentation in the agricultural domain. Comprehensive experiments are conducted to assess the efficacy of the ASA compared to the default SAM. The results show that the proposed model achieves significant improvements on all 12 agricultural segmentation tasks. Notably, the average Dice score improved by 41.48% on two coffee-leaf-disease segmentation tasks.

## 1. Introduction

The development and use of fundamental models in artificial intelligence have experienced rapid growth in recent years. These models are trained on large datasets for generalization across various tasks and domains. Large Language Models (LLMs) [1] have become widely adopted, making larger models a recent trend, for example, GPT4 [2] developed by OpenAI (2023). The Segmentation Anything Model (SAM) [3] is a powerful and flexible visual segmentation model that has gained a lot of attention for its ability to generate accurate and detailed segmentation masks based on user prompts. The model is trained on a large dataset of 11 million images and over 1 billion masks. It has a fast design and training process that enables it to adapt to new image distributions and tasks with zero samples. The SAM has shown its excellent segmentation capabilities in various scenarios, and it is bringing innovations to the field of image segmentation and computer vision.

In this paper, the SAM is introduced into agricultural segmentation tasks. In this context, it not only segments agricultural images with zero samples but also improves the accuracy and efficiency of crop disease identification and pest detection. In this case, the SAM can be a solution to agricultural segmentation problems caused by its specialization and insufficiency of available datasets, and provide scientific guidance and support for the agricultural industry. Crop disease segmentation is an important task in the field of agriculture, as it can prevent the spread and deterioration of diseases by detecting and locating them and taking necessary measures such as spraying pesticides and trimming leaves in time. This not only reduces pesticide usage and protects the ecosystem [4] but also improves crop yield and quality. Similarly, effective image segmentation of agricultural pests can also reduce their harm to crops [5]. For example, the level of damage and the effectiveness of pest control can be evaluated by segmenting out different types and numbers of pests to formulate a reasonable control program.

However, the SAM also has limitations like other basic models, especially in the case of a very wide range of applications of computer vision. Since the training data cannot cover all possible image types and the working scenarios are constantly changing, the SAM does not perform well in some agricultural image segmentation tasks and difficult scenarios. To demonstrate this, the SAM was tested on crop disease and pest segmentation tasks and it was found that the SAM does not “segment anything” well. This is because the training dataset of the SAM mainly consists of natural images, which often have clear edge information and differentiation, whereas agricultural images are taken in complex field environments, and have the following characteristics: (1) low contrast between the target and the background, which makes segmentation difficult, (2) the agricultural background is complex and varied, with foliage, soil, and rocks, (3) uneven illumination and shadows reduce the image quality and clarity, and (4) the targets have different morphologies, such as pests and diseases. These characteristics make it difficult for the SAM to adapt to the needs and characteristics of agricultural image segmentation, resulting in inaccurate or incomplete segmentation results. Therefore, the theoretical and practical problem this paper focuses on is as follows: How can these challenges be overcome so that the segmentation ability learned by the SAM on a massive dataset can benefit agricultural segmentation tasks?

Adaption [6,7,8] is an effective tool for fine-tuning basic, large vision models for downstream tasks, not only in NLP but also in computer vision. It requires only a small number of parameters to be learned (typically less than 5% of the total parameters) to allow efficient learning and faster updates while keeping most of the parameters frozen. It has also been shown that adaption methods work better than full fine-tuning because they avoid catastrophic forgetting and generalize better to out-of-domain scenarios, especially in low data states. It is believed that adaption is the most suitable technique for transferring the SAM to the agriculture domain. Therefore, a simple but effective adapter specifically designed for agricultural image segmentation is proposed. This adapter is a lightweight model that leverages internal and external knowledge to adapt to relatively fewer data and injects task-specific guidance information from the samples of that task. By using visual prompts to transfer information to the network, it can efficiently adapt the frozen large-scale base model to various agricultural image segmentation tasks with minimal additional trainable parameters. This adapter is integrated into the SAM to obtain an adapted model called the agricultural SAM adapter (ASA). To assess the performance of the ASA, a dataset containing 5464 images of agricultural pests and a dataset containing 1100 images of coffee-leaf diseases is collected. The pest dataset consists of 10 common pest types, whereas the coffee-leaf-disease dataset includes individual leaf images, as well as localized images of coffee trees. The extensive experimental results on 12 two-dimensional agricultural image segmentation tasks demonstrate that this approach significantly enhances the performance of the SAM in agricultural image segmentation tasks and compensates for its limitations in this context. The contributions of this paper can be summarized as follows:The SAM is applied to agricultural image segmentation for the first time, and a general method for agricultural image segmentation is developed.Two datasets are collected and labeled, which comprise 10 different pests and two coffee-leaf diseases, and experimental validation on these datasets is performed.The prompt content is designed according to the characteristics of the agricultural images, and the advantages of the ASA in the zero-shot segmentation of these images are demonstrated.A comprehensive evaluation of the proposed ASA through 12 agricultural image segmentation tasks is performed. The experimental results show that the ASA outperforms the SAM on the agricultural dataset.

The subsequent sections of this paper are structured as follows. Section 2 presents the research on adapters and agricultural image segmentation. In Section 3, the SAM and ASA are introduced. Section 4 provides a detailed description of the experimental setup employed and presents the obtained results. Finally, Section 5 concludes this paper.

## 2. Related Works

**Adapters.** The concept of adapters was initially introduced in the field of Natural Language Processing (NLP) [9] to fine-tune a large pre-trained model using a compact and scalable model for each specific downstream task. Stickland, Cooper, and Murray [10] explored multi-task approaches that shared a single BERT model with a small number of additional task-specific parameters using new adaptation modules, PALs, or “Projected Attention Layers” and obtained state-of-the-art results on the Recognizing Textual Entailment dataset. In the realm of computer vision, a method proposed by Facebook AI Research [11] achieved competitive results in object detection tasks with minimal adaptations for fine-tuning the ViT (Vision Transformer) architecture [12]. More recently, Chen et al. [13] designed a simple but powerful adapter for the ViT architecture for dense prediction tasks, and it demonstrated excellent performance in several downstream tasks, including object detection, instance segmentation, and semantic segmentation. Liu et al. [14] were inspired by the widely used pre-training and prompt-tuning protocols in NLP and proposed an EVP (Explicit Visual Prompting) technique that could efficiently combine explicit visual prompting with an adapter. This technique achieved state-of-the-art performance in low-level structure segmentation tasks. Additionally, Chen et al. presented the SAM adapter [15], achieving state-of-the-art results in camouflaged object detection and shadow detection tasks by incorporating domain-specific information and visual prompts into segmented networks. In this paper, the adapter approach is applied to the SAM to solve agricultural image segmentation tasks.

**Segmentation.** Image segmentation [16,17,18] is an important task in computer vision that aims to assign each pixel in an image to a specific semantic category. Image segmentation has a wide range of applications in fields such as agriculture, medicine, and remote sensing. However, traditional image segmentation methods usually rely on a large amount of annotated data for training models, which are difficult to obtain or costly in some specific domains. Therefore, exploring how to achieve image segmentation with few or no annotated data is both a challenging and valuable problem.

**Zero-Shot Segmentation.** Zero-shot segmentation is a method used for image segmentation that utilizes unlabeled data. This approach can segment objects of any class in an image based on various types of prompts, including points, bounding boxes, and text. The research field of zero-shot image segmentation has gained significant attention in recent years. For instance, Lüddecke and Ecker [19] proposed a backbone network based on CLIP and a Transformer-based decoder. This framework enables the generation of image segments by leveraging arbitrary text or image prompts, and its versatility in handling diverse segmentation tasks and generalizing to new queries has been demonstrated. Roy et al. [20] introduced SAM.MD, a zero-shot medical image segmentation method based on the SAM. The method effectively segments abdominal CT organs by utilizing point or bounding box prompts, and it has exhibited excellent performance across multiple medical datasets. Furthermore, research endeavors have explored the utilization of zero-shot image segmentation for more intricate tasks, including instance segmentation [21,22,23] and video segmentation [24]. Despite these advancements, the current state of zero-shot image segmentation methods in the field of agricultural image segmentation remains underexplored.

**Agricultural Image Segmentation.** Agricultural image segmentation is a crucial task with significant applications and practical relevance, as it aids agricultural producers in identifying and monitoring crop conditions and pests. This, in turn, improves the efficiency and quality of agricultural production. In recent years, deep learning methods, for example, the FCN (Fully Convolutional Network) [25], U-Net [26], and Mask R-CNN (Region-Based Convolutional Neural Networks) [23] have been widely used in agricultural image segmentation. These methods excel in automatically learning effective feature representations from data and are suitable for high-resolution and multi-class image segmentation tasks. Notably, deep learning techniques have achieved substantial performance advancements in crop disease and pest segmentation tasks, making them a mainstream approach. For instance, Ma et al. [27] used a DCNN (Deep Convolutional Neural Network) to develop a method for the automatic diagnosis of cucumber diseases. This method provided a scientific basis for farmers to apply pesticides appropriately. Another notable work by Esgario et al. [28] proposed a neural network-based multitasking system for the classification and severity estimation of coffee-leaf diseases. The system contains a suitable tool to assist experts and farmers in identifying and quantifying biotic stresses in coffee plantations. However, traditional deep learning approaches require extensive annotated data for training, which are scarce in the domain of agricultural image segmentation. Therefore, exploring how to extend zero-shot image segmentation methods to agricultural image segmentation presents a challenging and valuable research problem.

To address this research gap, the agricultural SAM adapter (ASA) is proposed, which is a fine-tuned version of the SAM customized for agricultural image segmentation, specifically designed for zero-shot segmentation with appropriate prompts. In this paper, experiments on 12 agricultural image segmentation tasks are conducted, and the results are shown in Figure 1, which demonstrates that the proposed approach significantly outperforms the original SAM in terms of performance and generalization capability.

## 3. Methods

### 3.1. Understanding the SAM’s Utility from an Agricultural Perspective

The SAM supports three main segmentation modes: fully automatic mode, bounding box mode, and point mode. In this study, an image of a weevil with a distinct contrast between the foreground and background is used as an example of segmentation using these three modes, and the corresponding results are presented in Figure 2. The fully automatic segmentation mode divided the entire image into several regions based on the intensity of the image (Figure 2b). However, in practical applications of agricultural image segmentation, this mode has limitations, as the focus is typically on the pests or crop diseases themselves. The bounding box-based segmentation model produced satisfactory results in weevil segmentation by providing only the top-left and bottom-right points (Figure 2c). Conversely, the point-based segmentation model (Figure 2(d-1–d-3)) initially resulted in segmenting only the pattern on the weevil’s back when a central foreground point was provided. To obtain the desired segmentation results for the weevil, two additional points at the front were added.

To assess the efficacy of the SAM in the segmentation of crop diseases, segmentation experiments were conducted, the results of which are depicted in Figure 3. It was observed that the point segmentation mode accurately separated the foreground after multiple iterations. Additionally, leaves with higher disease densities required an even greater number of iterations. Consequently, the point segmentation mode was not deemed suitable as a prompt for this experiment.

When employing the SAM for agricultural image segmentation, the fully automatic model yielded excessive region delineation, whereas the point model required multiple iterations of point prediction and correction to achieve the desired segmentation outcome. Conversely, the bounding box-based model permitted clear segmentation of both the foreground position and size without the need for multiple iterations to attain a satisfactory segmentation outcome. Hence, the bounding box-based segmentation model proved to be the more appropriate choice for agricultural image segmentation.

### 3.2. A Teardown Analysis of the SAM

Firstly, we begin by providing an overview of the SAM architecture. The SAM employs a converter-based architecture, known as a Transformer [29], which has exhibited impressive performance and flexibility in various domains such as NLP [9] and image recognition tasks [30]. The SAM consists of three key components: the image encoder, prompt encoder, and mask decoder. Figure 4a,b show the standard ViT block structure and mask decoder structure in the SAM, respectively.

**Image encoder.** The image encoder uses an MAE (Masked Autoencoder) [31] pre-trained ViT [12], minimally adapted to process high-resolution inputs [11]. The image encoder runs once per image and can be applied prior to prompting the model. Specifically, the ViT-H/16 variant is employed, which applies a 14 × 14 attention window and has four equally spaced global attention blocks. The output of the image encoder is a downsampled embedding of the input image, reduced by a factor of 16.

**Prompt encoder.** The prompt encoder employs different methodologies to encode the prompts based on the prompt type to convert the prompts into feature vectors. The prompts can be either sparse (points, bounding boxes, text) or dense (masks).

**Mask decoder.** The mask decoder consists of a Transformer-based two-layer decoder, which efficiently maps the image embedding, prompt embedding, and an output token to a mask. The decoder block uses self-attention and cross-attention in two directions to update all embeddings. After running two blocks, the image embeddings are upsampled, the output markers are mapped to the MLP (Multilayer Perceptron) of the dynamic linear classifier, and the mask’s probability is calculated for each image.

### 3.3. ASA: Dedicated Agricultural Image Segmentation Foundation Models

To adapt the SAM to agricultural image segmentation, it is necessary to choose a suitable user prompt and insert an adjusted network component. According to the analysis in Section 3.1, the bounding box prompt proves to be an appropriate selection for defining segmentation targets. The SAM’s network architecture comprises three primary components, the image encoder, prompt encoder, and mask decoder, with the flexibility to fine-tune any combination of these components. To minimize computational costs, the image encoder remains unaltered. The prompt encoder encodes information pertaining to the bounding box’s position and can be reused from the pre-trained bounding box encoder in the SAM; therefore, this component is also frozen. Ultimately, the designed adapter module is chosen to be inserted at a specific location in the mask decoder.

An adapter module is a straightforward and efficient approach to augmenting a network with new capabilities and information while preserving the original network parameters. Figure 4c shows the structure of the proposed adapter module, which encompasses two downscale layers, three ReLU activation layers, and two upscale layers. The initial downscale layer employs a fully connected layer to map the input embedding vector to a lower intermediate dimension, thereby reducing computational effort and parameter quantity. Subsequently, another downscale layer utilizes an additional fully connected layer to map the vectors, which have undergone the ReLU activation function, to the target dimension, ensuring compatibility with the inputs of the subsequent network layers. Meanwhile, the two upscale layers consist of fully connected layers, which map the vectors passing through the ReLU activation function back to the intermediate and original dimensions, respectively, to guarantee consistency in the network’s output. The activation layer employs a non-linear ReLU activation function to activate the vectors in the intermediate dimension, thereby incorporating non-linear information and enhancing expressiveness.

The proposed adapter is flexible, efficient, and generalizable. Its specific advantages are as follows: (1) the size of the intermediate dimensions can be adjusted according to different tasks and datasets, achieving different degrees of adaptation effects, (2) only linear transformations of the input and output vectors are required without complex self-attention operations, which can effectively reduce the amount of computations and parameters, thus improving the training and inference speed, and (3) the generalization ability of the SAM to agricultural image segmentation tasks can be improved so that it can better handle the diversity and complexity in agricultural scenes.

To align the SAM with the distinctive traits of agricultural image segmentation, three adapters were strategically frozen into the decoder architecture, as depicted in Figure 4d. The first adapter was implemented before the output of the multi-head attention layer in the prompt-to-image embedding and the residual connection, serving to regulate the interaction between the prompt and image embedding. Following the multi-headed attention layer. The second adapter was positioned after the output of the MLP layer, thereby enhancing the embedding of the MLP layer and improving its expressiveness. To connect the image embedding to the output of the prompt cross-attention layer, the third adapter was deployed. It is important to note that the adapters were exclusively deployed in the first of the two decoder blocks, balancing computational efficiency with performance gains. The second block and the mask prediction header were solely tailored to the original SAM data, ensuring the preservation of the original SAM’s capabilities.

## 4. Experiments and Results

### 4.1. Datasets

This study utilized three datasets to assess the model’s performance: the BRACOL dataset [28], the BRACOT dataset [32], and the Pest dataset [33]. The BRACOL and BRACOT datasets comprise images depicting coffee-leaf diseases captured in two distinct scenarios, whereas the Pest dataset comprises images of 10 prevalent agricultural pests. Table 1, provides the specific sample sizes of these datasets.

The BRACOL dataset comprises a total of 800 images of coffee-leaf diseases. These images were captured from the back (lower) side of the leaves, adhering to specific control conditions, and positioned on a white background. Several examples of these images are shown in Figure 5.

The BRACOT dataset comprises localized images of Arabica coffee trees exposed to various forms of biotic stress. The dataset encompasses a total of 300 images, capturing localized views of both healthy leaves and leaves affected by one or more types of biotic stress. Figure 6 illustrates some of the images in this dataset.

The Pest dataset comprises images of 10 prevalent agricultural pests, including the Cydia pomonella (codling moth), Gryllotalpa, leafhopper, locust (grasshopper), oriental fruit fly (Bactrocera dorsalis), Pieris rapae Linnaeus (Pieris rapae), snail, Spodoptera litura, stinkbug (Pentatomidae), and weevil. These pests are depicted in Figure 7. This dataset has been designed to facilitate research and applications in the field of pest detection and segmentation. It serves as a benchmark dataset for researchers and practitioners to develop and evaluate machine learning models. The dataset encompasses images featuring pests of diverse shapes, colors, and sizes, making it suitable for training and testing various algorithms in pest detection and segmentation across different scenarios.

### 4.2. Data Annotation and Preprocessing

To perform accurate analysis and processing of the aforementioned dataset, the images were annotated precisely to extract the regions of interest. An open source software named Labelme (v5.1.1) was utilized for pixel-level annotation of the datasets to label diseased areas and background areas in the coffee-leaf-disease dataset and pest areas against background areas for the Pest dataset. This pixel-level annotation provided valuable training data for image segmentation and the subsequent model’s design and evaluation.

To investigate the issue of image segmentation for agricultural pests and coffee-leaf diseases, a dataset consisting of 5464 images of agricultural pests and 1100 images of coffee-leaf diseases was constructed. Before training the network, a simple preprocessing of the dataset was performed to enhance the model’s performance and convergence speed. Specifically, the pixel values of all the images were normalized to a range of 0 to 255 and resized to a uniform size of 256 × 256 × 3.

### 4.3. Training Protocol

**Data partition.** The 10 types of pests in the Pest dataset were divided into two groups, each consisting of 5 types of pests. To adhere to the zero-shot segmentation setting, one group served as the training set, whereas the other group served as the test set. This process was repeated twice in reverse to assess the performance of the 10 types of pests using zero-shot segmentation. The dataset of coffee-leaf diseases was randomly divided into training and test sets with an 8:2 ratio. Segmentation targets with fewer than 100 pixels were disregarded to eliminate noise and interference.

**Model preset.** For the model, a pre-trained ViT-based model from the ImageNet dataset was utilized as the image encoder, which transformed input images into 1024-dimensional feature vectors. To improve computational efficiency, the normalized images were fed into the image encoder to obtain pre-computed image embeddings, which were then fed into the training network. During training, a bounding box prompt that followed the SAM’s idea was generated from the ground-truth mask with a random perturbation of 0–20 pixels, after which the ASA generated outputs based on the prompt. The unweighted sum of the Dice loss and cross-entropy loss was used as the loss function, as it has shown robustness in various segmentation tasks.

**Hyperparameters.** The network was optimized using the Adam optimizer, with an initial learning rate of $1\times 10^{-5}$ and a 0 weight decay. The batch size was set to 8. The specific epochs used in the different datasets can be seen in Table 2. The time span of each training is also recorded. The loss curves, plotted for the crop disease segmentation and pest segmentation tasks for the ASA, are shown in Figure 8. The pre-trained model generated by the experiment can be downloaded from https://drive.google.com/file/d/1CRyX2CZQMT7sA9J7ODoUSoCV4g1AUXMb/view?usp=sharing (accessed on 9 September 2023).

**Training environments.** All the data preprocessing, model training, and evaluating tasks were run on a physical machine equipped with an Intel Xeon Platinum 8380 Processor, two NVIDIA A30 Tensor Core GPUs, and 256 GB of memory.

### 4.4. Evaluation Metrics

The quality of the segmentation results was assessed using the Dice coefficient (Dice) and intersection over union (IoU) as the evaluation metrics [34], which are commonly utilized in segmentation analysis. These metrics measure the degree of regional overlap and consistency between the segmentation results and the true value mask.

**Dice Score (Coefficient).** The Dice coefficient measures the degree of overlap between the predicted mask and the ground truth. It corresponds to the ratio of twice the overlap area to the total number of pixels in the ground-truth image and the segmentation output.

**Mean Intersection Over Union (mIoU).** The intersection over union (IoU) is defined as the overlap (intersection) area divided by the union area between the ground truth and predicted segmentation. The mean IoU (mIoU) is widely used for multi-class semantic segmentation and is calculated by averaging the IoUs calculated for each class. It is a much more effective measure than pixel accuracy and does not suffer from the class imbalance issue.

### 4.5. Evaluation Results in the Agricultural Image Segmentation Task

A quantitative comparison was conducted between the pre-trained SAM (ViT-B) and the ASA in the agricultural image segmentation task. The results of this comparison are presented in Table 3 and Table 4. The ASA exhibited significantly superior performance compared to the SAM (ViT-B) in the agricultural image segmentation task, as demonstrated in the tables. Specifically, on the coffee-leaf dataset, the ASA achieved an average Dice improvement of 41.48% and an average IoU improvement of 36.63%. On the pest dataset, the average Dice improvement reached 9.91%, accompanied by an average IoU improvement of 11.19%.

To demonstrate the comprehensive advantages of our approach, additional examples of comparative segmentation results are provided in Figure 9. The pre-trained SAM exhibited significant shortcomings in agricultural image segmentation, particularly in terms of over-segmentation and inaccurate segmentation. These issues arose when segmenting images with low contrast or shadow interference between the target and the background. Specifically, when segmenting leaf diseases, the SAM incorrectly identified the edges or background of the leaf as diseases, instead of accurately detecting the diseased areas. Similarly, for pests with simple morphologies, high contrast, and clear outlines (e.g., leafhopper, Pieris rapae Linnaeus, snail, Spodoptera litur), the SAM achieved better segmentation results. However, for pests with complex morphologies, low contrast, and blurred outlines (e.g., locust, Gryllotalpa, oriental fruit fly, stinkbug), the performance was inadequate, and accurate segmentation of the pests’ heads, antennae, wings, etc., was not achieved. Moreover, the SAM frequently failed to generate satisfactory segmentation results for pest images with unclear boundaries, shadows, or target colors that were similar to the background. Conversely, the proposed model, after targeted training, demonstrated significant advantages in agricultural image segmentation tasks. It enabled effective segmentation of leaf-disease regions and substantially enhanced the accuracy and integrity of pest regions. Unlike the SAM, the ASA exhibited greater adaptability to the characteristics and challenges in agricultural images and avoided over-segmentation and inaccuracies.

Figure 10 and Figure 11 depict box plots illustrating the experimental results to compare the performance of the ASA and SAM. The box plots indicate that the ASA generated shorter boxes, which suggests a more concentrated distribution and higher stability and consistency of the experimental results. The medians depicted on the box plots consistently show that the ASA outperformed the SAM by achieving higher values across all datasets, which implies better prediction results. The quartiles represented on the box plots demonstrate that the ASA achieved a narrower range of values across all datasets, indicating lower volatility and closer proximity of the experimental results to the median level. The upper and lower edges of the box plots illustrate that the SAM achieved a longer range of extreme values across all datasets, which indicates a higher variation and reflects a relatively larger variance and standard deviation of the overall datasets.

This paper proposes an adaptive segmentation approach (ASA) for agricultural image segmentation and compares it with a pre-trained segmentation model (SAM) through experiments. The experimental results demonstrate that the proposed model exhibits significant advantages in both leaf-disease and pest segmentation tasks, delivering efficient, stable, and consistent performance. Conversely, the SAM suffers from issues such as over-segmentation, inaccurate segmentation, and reduced effectiveness in handling images with low contrast or shadow interference. The analysis of the box plots additionally confirms the superior performance of the ASA compared to the SAM across all datasets, as evidenced by higher median values, narrower interquartile distances, and more localized top and bottom edges.

## 5. Conclusions

The proposed agricultural SAM adapter (ASA) is a simple but effective model that adapts the powerful general-purpose Segment Anything Model (SAM) to the field of agricultural image segmentation. The proposed model not only achieves zero sample agricultural image segmentation but also significantly improves the performance in both agricultural pest segmentation and leaf-disease segmentation tasks. Experiments were conducted on 2 coffee-leaf-disease segmentation tasks and 10 pest segmentation tasks, achieving state-of-the-art results. These results demonstrate the effectiveness of the proposed approach and the transferability of a generic segmentation model to agricultural applications.

## Figures and Tables

**Figure 1 sensors-23-07884-f001:**
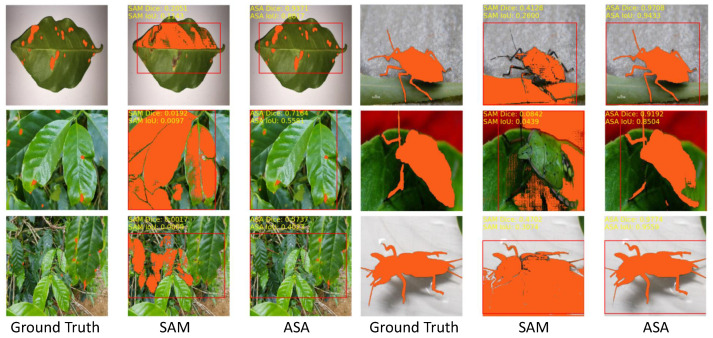
Visualized examples of the pre-trained SAM and ASA segmentation results. The ASA significantly improves segmentation performance in agricultural image segmentation tasks.

**Figure 2 sensors-23-07884-f002:**
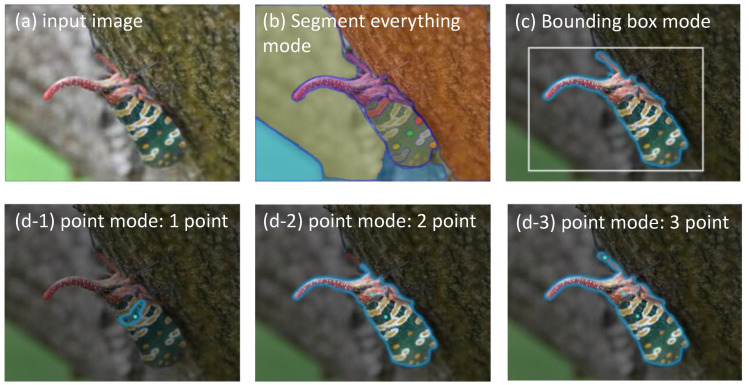
Segmentation results of the SAM based on different segmentation modes.

**Figure 3 sensors-23-07884-f003:**
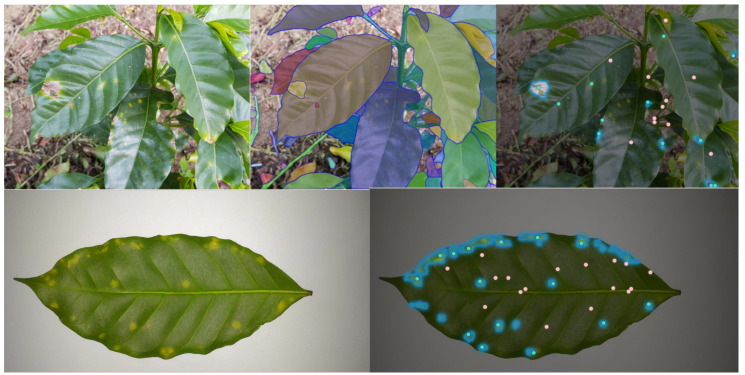
Segmentation results of the SAM based on the point segmentation model. Light green dots stand for the foreground while pink dots stand for the background.

**Figure 4 sensors-23-07884-f004:**
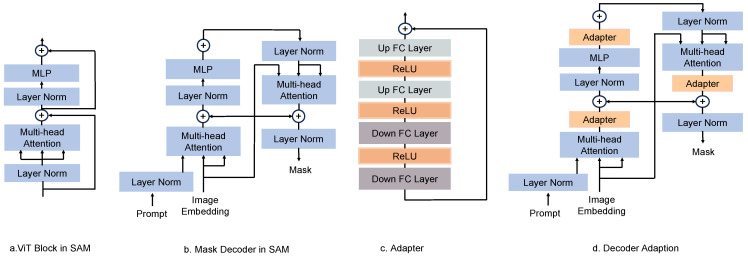
ASA architecture: By adding adapters to the SAM for agricultural image segmentation, the image encoder and prompt encoder are frozen and adapters are only inserted in the mask decoder. Note that the prompt’s self-attention module and mask prediction head are not shown in the decoder. (**a**) Standard ViT block structure in the SAM. (**b**) Mask decoder structure for the SAM. (**c**) The proposed adapter. (**d**) Mask decoder inserted into the adapter.

**Figure 5 sensors-23-07884-f005:**
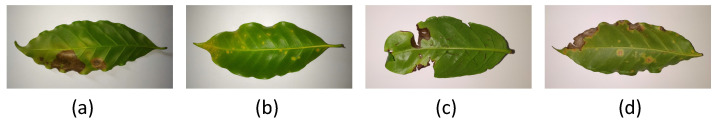
Examples of coffee leaves affected by different biotic stresses: leaf miner (**a**), rust (**b**), brown leaf spot (**c**), and Cercospora leaf spot (**d**).

**Figure 6 sensors-23-07884-f006:**
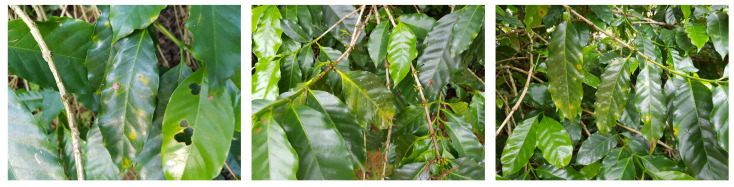
Several example images from the BRACOT dataset.

**Figure 7 sensors-23-07884-f007:**
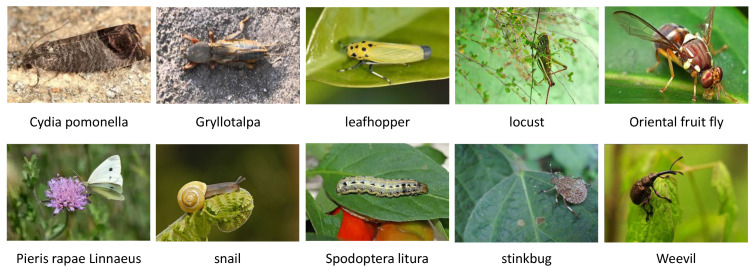
Several example images from the Pest dataset.

**Figure 8 sensors-23-07884-f008:**
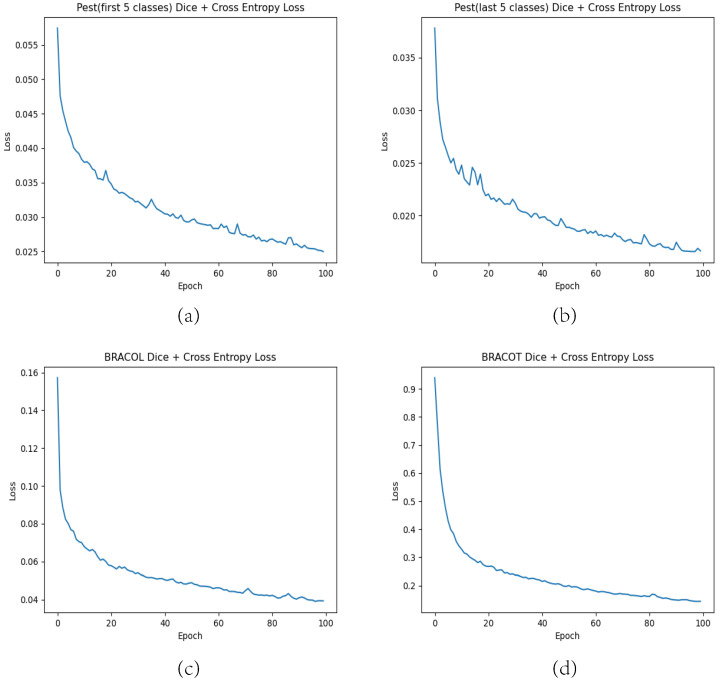
Loss curves for different segmentation tasks. (**a**) Training curves for the first 5 pests in the Pest dataset. (**b**) Training curves for the last 5 pests in the Pest dataset. (**c**) Training curves for the BRACOL dataset. (**d**) Training curves for the BRACOT dataset.

**Figure 9 sensors-23-07884-f009:**
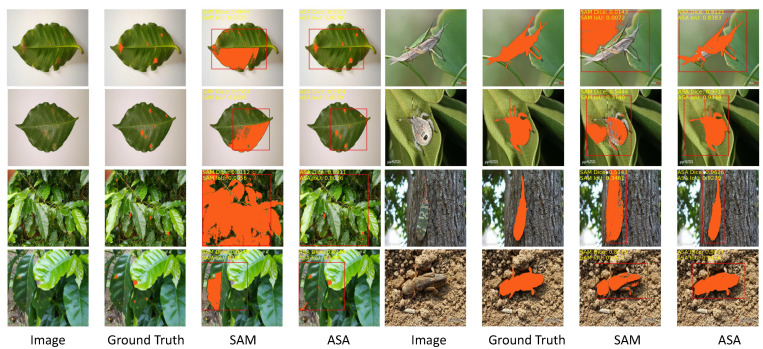
More examples of the pre-trained SAM and ASA in agricultural image segmentation tasks.

**Figure 10 sensors-23-07884-f010:**
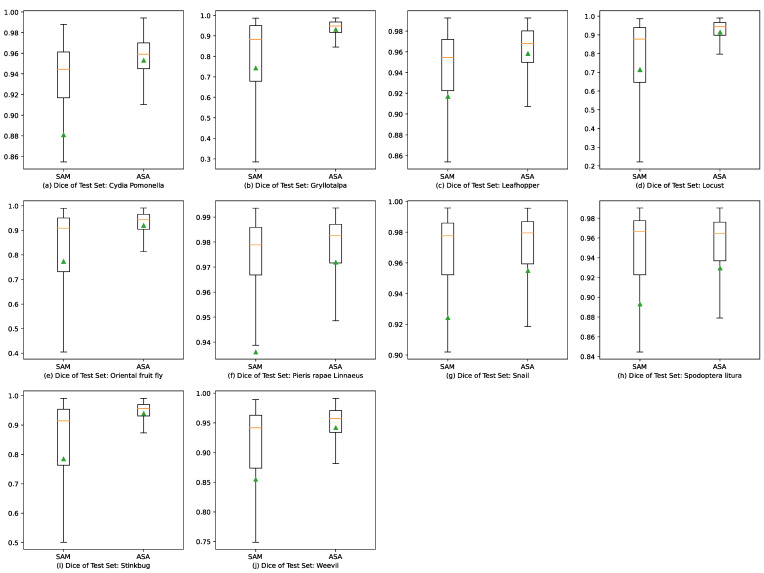
Box plots of Dice results for the pre-trained SAM and ASA in 10 agricultural pest image segmentation tasks. The triangle stands for the mean value of the test Dice. The orange lines through the boxes stand for the median numbers of the accuracies. And the green triangles stand for the average of the accuracies.

**Figure 11 sensors-23-07884-f011:**
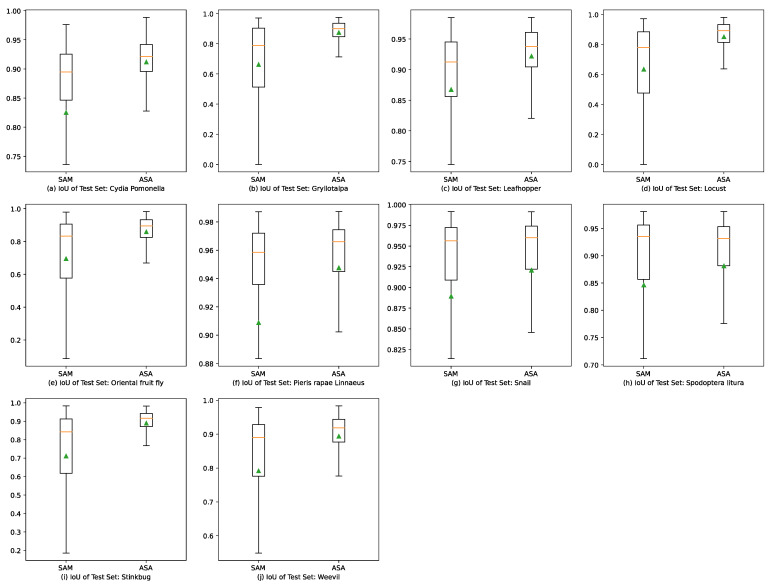
Box plots of IoU results for the pre-trained SAM and ASA in 10 agricultural pest image segmentation tasks. The triangle stands for the mean value of the test IoU. The orange lines through the boxes stand for the median numbers of the accuracies. And the green triangles stand for the average of the accuracies.

**Table 1 sensors-23-07884-t001:** Brief description of the sample size of the dataset.

Dataset	Sample Size
Cydia pomonella	376
Gryllotalpa	469
Leafhopper	423
Locust	664
Oriental fruit fly	455
Pieris rapae Linnaeus	566
Snail	904
Spodoptera litura	394
Stinkbug	661
Weevil	552
BRACOL dataset	800
BRACOT dataset	300

**Table 2 sensors-23-07884-t002:** Details of training parameters for different datasets. Note that the iters for the speed refers to the number of iterations of each epoch and was obtained from the Image_amount/Batch_size.

Dataset	Epochs	Time Span (s)	Speed (≈)
Pest (first 5 classes)	100	1478	20 iters/s
Pest (last 5 classes)	100	2026	19 iters/s
BRACOL	100	390	20 iters/s
BRACOT	100	150	20 iters/s

**Table 3 sensors-23-07884-t003:** Comparison of the performance of the ASA and SAM in coffee-leaf-disease segmentation tasks. The ASA achieved significant and consistent improvements across all tasks. The improvements of ASA are bolded in the table.

Segmentation Target	mDice (%)			mIoU (%)		
	SAM	ASA	Improve	SAM	ASA	Improve
BRACOL dataset	47.54	90.00	**42.46**	41.72	83.96	**42.24**
BRACOT dataset	6.08	46.58	**40.50**	4.18	35.19	**31.01**
Average	26.81	68.29	**41.48**	22.95	59.58	**36.63**

**Table 4 sensors-23-07884-t004:** Comparison of the performance of the ASA and SAM in agricultural pest image segmentation tasks. The ASA achieved significant and consistent improvements across all tasks. The improvements of ASA are bolded in the table.

Segmentation Target	mDice (%)			mIoU (%)		
	SAM	ASA	Improve	SAM	ASA	Improve
Cydia pomonella	88.11	95.32	**7.21**	82.52	91.22	**8.7**
Gryllotalpa	74.38	93.06	**18.68**	66.34	87.51	**21.17**
Leafhopper	91.71	95.84	**4.13**	86.74	92.20	**5.46**
Locust	71.42	91.47	**20.05**	63.58	85.16	**21.58**
Oriental fruit fly	77.36	92.02	**14.66**	69.58	85.96	**16.38**
Pieris rapae Linnaeus	93.61	97.19	**3.58**	90.90	94.76	**3.86**
Snail	92.44	95.50	**3.06**	88.96	92.10	**3.14**
Spodoptera litura	89.33	92.95	**3.62**	84.64	88.17	**3.53**
Stinkbug	78.53	93.97	**15.44**	71.20	89.08	**17.88**
Weevil	85.49	94.20	**8.71**	79.21	89.45	**10.24**
Average	84.24	94.15	**9.91**	78.37	89.56	**11.19**

## Data Availability

Publicly available datasets were analyzed in this study. These data can be downloaded from https://doi.org/10.17632/yy2k5y8mxg.1 and https://data.mendeley.com/datasets/pmkbyjpf6k/1 and https://sejonguniversity-my.sharepoint.com/personal/hanxiang_sju_ac_kr/_layouts/15/onedrive.aspx?id=%2Fpersonal%2Fhanxiang%5Fsju%5Fac%5Fkr%2FDocuments%2FPest%20dataset&ga=1 (accessed on 12 July 2023).

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
