# Peer review of "Enhancing Agricultural Image Segmentation with an Agricultural Segment Anything Model Adapter"

_sensors, 2023, doi:10.3390/s23187884_

Round 1
Reviewer 1 Report
The authors aim to customize the Segment Anything Model (SAM) to the task of agricultural image segmentation. As they report, this is the first attempt of this kind, and this is the novelty of the study. The customization of SAM involves a user prompt, which is being selected as user-selected rectangular bounding box. With these modifications the method show better performance than the baseline. Numerical experiments are carried out with publicly available databases. The text is clear, all necessary parts are present. Although not a breakthrough, the paper presents a quality study in the certain problem domain. It can be published after some clarifications, given below.
It is necessary to present more details on method performance:
a) number of tunable parameters used in the method (and in baseline method, for comparison);
b) since the method is based on training, it is advisable to present a speed of training (saturation graph or something like that);
c) necessary equipment and time of processing should be described, especially when the user intervention is involved in the processing.
Author Response
Comments:
a) number of tunable parameters used in the method (and in baseline method, for comparison);
b) since the method is based on training, it is advisable to present a speed of training (saturation graph or something like that);
c) necessary equipment and time of processing should be described, especially when the user intervention is involved in the processing.
Responses:
a) The hyper-parameters used of training the proposed model has been added into the revised manuscript. Note that the comparison, SAM model, is pretrained, there is no need for us to train it in the experiments. So there is no hyper-parameters for the SAM.
b) The training speed and the convergence curve of losses has been added into the revised manuscript.
c) The equipment especially the hardware of running the experiments have been recorded in the revised manuscript.
Changes Made:
a) The changes of the response to the comment are in section 4.3. To be specific, There is a paragraph starts with Hyper-parameters.
b) The training speed and the convergence curve of losses are shown in table 2 and figure 8.
c) The equipment is described in section 4.3 - Training environments. The time of processing is shown in table 2.
Reviewer 2 Report
This paper considered an adaptive segmentation approach for agricultural image segmentation and compares it with a pre-trained segmentation model through experiments. THowever, the novelty is trial and the contribution is weak. What is the difficulty of introducing the segmentation tasks and consistent performance? It is unclear for how to solve the difficulty of the problem. What knowledge gap does your proposed method covers?
The writing is too poor to follow. The Abstract is too long and does not focus the main contribution. Many notations are used without introduction. Many abbreviations are dulplicated too many times. THe Introduction section should not put on a figure. The upper case letters wrongly used in many places of the titles of references. The algorithms and the frame figures are unclear.
Besides, English grammar, spelling,
and sentence structure are too poor to be understood, such as two many subjcetive terms "we", "our" are involved.
The following related adaptive analysis and image processing references should be cited to highlight the motivation.
[1] Injected infrared and visible image fusion via L1 decomposition model and guided filtering, IEEE Transactions on Computational Imaging, 8: 162-173, 2022.
[3] Adaptive fuzzy fault-tolerant control of uncertain Euler-Lagrange systems with process faults, IEEE Transactions on Fuzzy Systems, 2020, 28(10): 2619-2630.
[4] Fuzzy adaptive output feedback control of uncertain nonlinear systems with prescribed performance, IEEE Transactions on Cybernetics, 2018, 48(5): 1342-1354.
This paper considered an adaptive segmentation approach for agricultural image segmentation and compares it with a pre-trained segmentation model through experiments. THowever, the novelty is trial and the contribution is weak. What is the difficulty of introducing the segmentation tasks and consistent performance? It is unclear for how to solve the difficulty of the problem. What knowledge gap does your proposed method covers?
The writing is too poor to follow. The Abstract is too long and does not focus the main contribution. Many notations are used without introduction. Many abbreviations are dulplicated too many times. THe Introduction section should not put on a figure. The upper case letters wrongly used in many places of the titles of references. The algorithms and the frame figures are unclear.
Besides, English grammar, spelling,
and sentence structure are too poor to be understood, such as two many subjcetive terms "we", "our" are involved.
The following related adaptive analysis and image processing references should be cited to highlight the motivation.
[1] Injected infrared and visible image fusion via L1 decomposition model and guided filtering, IEEE Transactions on Computational Imaging, 8: 162-173, 2022.
[3] Adaptive fuzzy fault-tolerant control of uncertain Euler-Lagrange systems with process faults, IEEE Transactions on Fuzzy Systems, 2020, 28(10): 2619-2630.
[4] Fuzzy adaptive output feedback control of uncertain nonlinear systems with prescribed performance, IEEE Transactions on Cybernetics, 2018, 48(5): 1342-1354.
Author Response
Comment:
a) Limited Novelty: The approach's novelty and contribution are weak.
b) Writing Issues:
Abstract is long and unfocused.
Notations lack introduction.
Abbreviations duplication.
Misuse of uppercase letters in references.Figure in introduction.
Frame of algorithm and model is unclear.
c) Language Concerns:
grammar, spelling Too many subjective terms "we", "our" are involved.
d) References:
The following related adaptive analysis and image processing references should be cited to highlight the motivation.
[1] Injected infrared and visible image fusion via L1 decomposition model and guided filtering, IEEE Transactions on Computational Imaging, 8: 162-173, 2022.
[3] Adaptive fuzzy fault-tolerant control of uncertain Euler-Lagrange systems with process faults, IEEE Transactions on Fuzzy Systems, 2020, 28(10): 2619-2630.
[4] Fuzzy adaptive output feedback control of uncertain nonlinear systems with
prescribed performance, IEEE Transactions on Cybernetics, 2018, 48(5): 1342-1354.
Response:
a) More researches to support the novelty have been made and the results are presented in the introduction, the changes are highlighted.
b)We've reconsidered the arrangement of the abstract. And the revised abstract is presented. We provide further explanation of the evaluation indicators.
A large number of abbreviations have been removed from the revised draft and alternative words have been used.
All references have been reorganized and cited.
The figure in the introduction was moved to the end of the related work.
The description of the algorithm is modified and supplemented in the revised version, and the modeling framework is redrawn with new SAM-related models. Notably, we have swapped the original sections 3.1 and 3.2 to improve the coherence of the article.
c) Thank you for the reminder. I've changed the grammar and spelling in the revised draft. In the revised draft, a number of changes have been made to the subjective terms "we" and "our".
d) You give three papers that are very inspiring. They have been cited in the revised draft.
Changes Made:
a) The novelty and contribution of the study are reintroduced in the introduction.
Specifically, agricultural images are captured in more complex environments than natural images, with more specific segmentation tasks and fewer specialized datasets available. In order to solve this problem, the expertise in the field of agriculture is integrated into these segmentation model through adapter technology, and the ability of the large model is utilized to achieve zero-shot segmentation and enhance the segmentation effect.
b) We have made changes to the Abstract section and have highlighted the changes.
In Section 4.4 we explain in detail the two evaluation metrics used, Dice and IoU. A large number of abbreviations appearing in section 4.5 have been changed. In addition, individual abbreviations have been replaced in the text. There is a paragraph starts with "Figure 10. and Figure 11."
We have recited all references, See references for details.
The figures in the introduction have been moved to the last paragraph of Related work. It has been highlighted in the appropriate place.
The algorithmic framework of Figure 4 has been redrawn, and the corresponding framework diagram for SAM has been added. The order of section 3.1 and 3.2 was switched. A description of the SAM model is now added in section 3.2. A description of our algorithm has been added to section 3.3, which is labeled with highlights.
c) We have deleted "we" and replaced "our" in the text, e.g., "The method", "The model". We have also highlighted and annotated several parts of the revised manuscript.
d) The papers are cited in references 7, 8 and 17.
I would like to express my heartfelt gratitude for your thorough review and insightful comments on my manuscript. Your expertise and attention to detail have greatly contributed to the improvement of my research work. I truly appreciate the time and effort you have dedicated to providing constructive feedback.
Reviewer 3 Report
The purpose of manuscript is to extend SAM’s (Segment Anything Model) robust segmentation capabilities in the realm of agricultural image segmentation. The article has a practical importance for agricultural sector. The structure of manuscript is adequate.
However, there are some shortcomings:
1. Explain in a legend the mean of bars from figures.
2. Give more information for the role of this study in agriculture
3. The results are less discussed in relation to the specific literature.
4. Include scientific name for plants, pests, diseases.
Author Response
Comment:
a) Explain in a legend the mean of bars from figures.
b) Give more information for the role of this study in agriculture.
c) The results are less discussed in relation to the specific literature.
d) Include scientific name for plants, pests, diseases.
Response:
a) Re-plotted Figures 10 and 11 and added mean data.
b) Provide more of the role of this study in agriculture in the introduction of the revised manuscript.
c) A discussion of specific literature was added and the results were supplemented.
d) Relevant scientific names of pests and diseases have been added accordingly.
Changes Made:
a) Figures 10 and 11 in section 4.5 were redrawn to increase the mean values.
b) In the introduction, the role of this research in agriculture is added. Specifically, the second paragraph of the introduction, beginning with the words "We try".
c) A discussion of specific literature has been added to paragraphs 1 and 4 of the related work, and the results have been supplemented. Specifics are highlighted.
d) The addition of scientific names of plant diseases and pests has been added to the introduction to the dataset section 4.1. Plant diseases are shown in Figure 5 and pest scientific names are added in the fourth paragraph of section 4.1. All modifications have been highlighted.
Your valuable suggestions and recommendations have helped me refine my ideas and strengthen the overall quality of the paper. Your expertise in the field has been instrumental in guiding me towards a more comprehensive and rigorous analysis. I am genuinely grateful for your professional insights and the meticulousness with which you reviewed my work.
Round 2
Reviewer 2 Report
The authors revised the paper according to my comments. However, the novelty is still trial and has not theoretical values. It is unclear for how to solve the difficulty of the problem. Important and recent references are insufficient. The writing is poor, since the uppercase and lowercase letters are wrongly used and subjective expression. The Abstract and Conclusion are badly and tediously organized. The grammar errors own even in colored text. The writing is not a proper formulation and the reader cannot know what it is. Besides, English grammar, spelling,
and sentence structure are too poor to be understood, such as two many subjective terms "we", "our" are involved. e.g. "data is", "such as e.g.", "could be", "may ..." "would be".
The following recently and related adaptive analysis and image processing references should be cited to highlight the motivation.
[1] Fuzzy adaptive output feedback control of uncertain nonlinear systems with prescribed performance, IEEE Transactions on Cybernetics, 2018, 48(5): 1342-1354.
[2] Injected infrared and visible image fusion via L1 decomposition model and guided filtering, IEEE Transactions on Computational Imaging, 8: 162-173, 2022.
[3] Applications of fractional operator in image processing and stability of control systems, Fractal Fract, 7(5):359, 2023.
The authors revised the paper according to my comments. However, the novelty is still trial and has not theoretical values. It is unclear for how to solve the difficulty of the problem. Important and recent references are insufficient. The writing is poor, since the uppercase and lowercase letters are wrongly used and subjective expression. The Abstract and Conclusion are badly and tediously organized. The grammar errors own even in colored text. The writing is not a proper formulation and the reader cannot know what it is. Besides, English grammar, spelling,
and sentence structure are too poor to be understood, such as two many subjective terms "we", "our" are involved. e.g. "data is", "such as e.g.", "could be", "may ..." "would be".
The following recently and related adaptive analysis and image processing references should be cited to highlight the motivation.
[1] Fuzzy adaptive output feedback control of uncertain nonlinear systems with prescribed performance, IEEE Transactions on Cybernetics, 2018, 48(5): 1342-1354.
[2] Injected infrared and visible image fusion via L1 decomposition model and guided filtering, IEEE Transactions on Computational Imaging, 8: 162-173, 2022.
Author Response
I would like to express my sincere gratitude for the your insightful comments, which have significantly contributed to improving the quality and clarity of my work. I am pleased to outline the major changes made in light of the reviewers' comments:
Reviewer 2:
Comment 1: The novelty is still trial and has not theoretical values. It is unclear for how to solve
the difficulty of the problem.
Response 1: Thank you very much for your feedback and for reviewing our work. We greatly appreciate your concern regarding the novelty and theoretical value of the research. The primary objective of our study is to explore and validate new methods or ideas and observe their practical effectiveness. While the research is still in the experimental stage, we believe that valuable data and insights can be gathered through practical experiments and analysis.
The novelty, theoretical value and solution of this paper are explained below: The novelty of the article lies in proposing a simple yet effective agricultural image segmentation adapter (ASA) that adapts a generalized segmentation model (SAM) to the agricultural domain and achieves zero-sample agricultural image segmentation. The theoretical value lies in exploring how the adapter technique can be used to migrate large-scale pre-trained models to specific domains and tasks while retaining the parameters and capabilities of the original models, improving the efficiency and generalizability of the models. The challenge lies in choosing the appropriate user prompts and adapter locations, as well as how to adjust the intermediate dimensions and learning rates of the adapters to achieve the best segmentation results. The solution lies in choosing the bounding box prompt as the definition of the segmentation target according to the characteristics of agricultural images and inserting three adapter modules into the mask decoder to optimize the model's performance on the task of agricultural image segmentation by fine-tuning a small number of parameters.
Comment 2: Important and recent references are insufficient.
Response 2: Thanks to the reviewers for this suggestion, we have added new references as 7, 8, 17. and have highlighted. The following recent and relevant adaptive analysis and image processing references have been cited.
[1] Fuzzy adaptive output feedback control of uncertain nonlinear systems with prescribed
performance, IEEE Transactions on Cybernetics, 2018, 48(5): 1342-1354.
[2] Injected infrared and visible image fusion via L1 decomposition model and guided filtering,
IEEE Transactions on Computational Imaging, 8: 162-173, 2022.
[3] Applications of fractional operator in image processing and stability of control systems,
Fractal Fract, 7(5):359, 2023.
Comment 3: The writing is poor, since the uppercase and lowercase letters are wrongly used and subjective expression. The grammar errors own even in colored text. The writing is not a proper formulation and the reader cannot know what it is. Besides, English grammar, spelling, and sentence structure are too poor to be understood, such as two many subjective terms "we",
"our" are involved. e.g. "data is", "such as e.g.", "could be", "may ..." "would be".
Response 4: Thanks for the advice. We have heavily revised and highlighted grammar and spelling in the text, including grammar, spelling, capitalization, and the use of "such as e.g."、"could be"、"may ..."、"would be".
Comment 4: The Abstract and Conclusion are badly and tediously organized.
Response 4: Thanks to the reviewer for pointing this out, we do have a problem with lengthy abstracts. We have simplified the abstract in the revised manuscript. The logic of the abstract and conclusion has been reorganized. Changes can be seen in the Abstract and Conclusion sections and are highlighted.
A revised manuscript based on the review is attached.

Reviewer 3 Report
The authors generally responded well to my comments. The article is much better.
Author Response
Thank you for providing me with the reviewers’ comments on my paper. I am very happy to see that you think that I have responded well to your suggestions, and that my paper has improved a lot. I sincerely appreciate the your professional level and meticulous work, which have been very helpful for improving the quality of my paper.
Thank you again for your attention and help on my paper. I look forward to your final approval of my revised paper.